# Neutrophil extracellular traps in the animal model of adenine-induced chronic kidney disease

**Alexandra Gaál Kovalčíková**[1,2], **Gabriela Forraiová**[2], **Nelia Korotushak**[2], **Veronika Borbélyová**[2], **Diana Vavrincová-Yaghi**[3], **Peter Vavrinec**[3], **Ľudmila Podracká**[1], **Peter Celec** [2,4]*

1 Department of Pediatrics, National Institute of Children's Diseases and Faculty of Medicine, Comenius University, Bratislava, Slovakia, 2 Institute of Molecular Biomedicine, Faculty of Medicine, Comenius University, Bratislava, Slovakia, 3 Department of Pharmacology and Toxicology, Faculty of Pharmacy, Comenius University, Bratislava, Slovakia, 4 Institute of Pathophysiology, Faculty of Medicine, Comenius University, Bratislava, Slovakia

* petercelec@gmail.com

## Abstract

Neutrophil extracellular traps (NETs) are suggested to play a role in chronic kidney disease (CKD). Whether they are indeed involved in the pathogenesis of animal models of CKD has not been proved. This study tested the hypothesis that a genetic deficiency of peptidylarginine deiminase 4 (PAD4) – a regulator of NETs production will protect mice from developing adenine-induced nephropathy as a model of CKD. Adult male Padi4$^{-/-}$ mice and their wild-type counterparts received 100 mg/kg adenine or saline i.p. daily for 14 days. Markers of renal function and NETs-related biomarkers were assessed during the CKD induction in plasma and urine. In wild-type mice, adenine injections decreased body weight by 20% and increased plasma creatinine (twofold), neutrophil gelatinase-associated lipocalin (21-fold), and neutrophil elastase (4-fold). No significant differences were found between the Padi4$^{-/-}$ and wild-type mice except the earlier increase in plasma creatinine in Padi4$^{-/-}$ mice (day 3 vs. day 7). Analyses of neutrophil elastase and myeloperoxidase in plasma and urine suggest that neutrophils are activated in adenine-induced nephropathy, but the production of NETs seems not to be directly involved in the pathogenesis of this CKD model. Further studies should clear the role of NETs in other kidney disease models with clinical relevance.

## Introduction

Neutrophils, as the main effector cells of the innate immunity, employ multiple mechanisms to execute their defensive functions. Among the most recently identified processes is the release of mesh-like structures called neutrophil extracellular traps (NETs). NETs consist of decondensed chromatin scaffolds decorated with histones and a repertoire of granule-derived effector molecules, including neutrophil

**Data availability statement:** All relevant data are available from the Figshare database https://figshare.com/s/31b1de88e738ffb7616d.

**Funding:** This work was supported by the Slovak Research and Development Agency (grants APVV-23-0399 to DVY and APVV-22-0554 to PC) and by the Grant Agency of the Ministry of Education, Research, Development and Youth of the Slovak Republic (grants VEGA 1/0801/25 to AGK, VEGA 1/0297/26 to PV, and VEGA 1/0513/24 to DVY). No additional internal or external funding was received for this study. The funders had no role in study design, data collection and analysis, decision to publish, or preparation of the manuscript.

**Competing interests:** The authors have declared that no competing interests exist.

elastase (NE), myeloperoxidase (MPO), gelatinase, and other antimicrobial peptides [1]. Histone citrullination by protein-arginine deiminase type 4 (PAD4) initiates the decondensation of chromatin, which precedes NETs release [2]. Besides the pivotal role of NETs in infections and pathogen clearance, they also cause inflammation, tissue damage and increase the risk of autoimmune diseases [3]. In sterile inflammation, neutrophils recognize molecules released from damaged and dying cells as damage-associated molecular patterns (DAMPs), leading to an immune response. Extravasation of neutrophils into damaged tissues with subsequent NETs production amplifies inflammation and tissue damage, resulting in a vicious cycle relevant for several different pathologies [4].

Dysregulation of the crosstalk between kidneys and the immune system is one of the underlying mechanisms of numerous kidney disorders [5], with NETs implicated in both acute kidney injury (AKI) [6–9] and chronic kidney disease (CKD) [10–13]. NETs-associated components such as extracellular DNA (ecDNA), histones, or granular proteins may act as antigens for autoantibody production in diseases including lupus nephritis or ANCA-vasculitis and thus may be directly involved in the etiology of CKD. Studies have demonstrated strong correlations between circulating NETs levels, their deposition in the kidneys, and the severity of clinical signs in CKD patients [10,14–16]. Furthermore, the secondary inflammation resulting from CKD can cause immune dysregulation, excessive NETs formation, and impaired NETs clearance, creating a detrimental cycle. The accumulation of cytotoxic NETs components can induce apoptosis of renal endothelial cells, degrade the basement membrane, and promote endothelial-to-mesenchymal transition, ultimately leading to renal fibrosis [17,18]. NETs also promote thrombosis [19,20], and in CKD patients uremic conditions dysregulate neutrophil activity, further increasing NETs formation and vascular complications. Elevated NETs-markers concentrations predict poor cardiovascular outcomes and mortality in CKD patients [21–23] and may also be involved in complications related to kidney transplant rejection [24,25]. Thus, NETs are associated with CKD, however, the causality and direction of this association remain complex.

Crystals of various origins can cause a wide spectrum of disorders, including kidney diseases that involve metabolic dysregulation and robust inflammatory responses, contributing to tissue damage. The mechanism and severity of the immune response in crystalline nephropathy depend on the source, size, or shape of the crystals [26], with NETs formation being identified as a significant contributor to its pathogenesis [27,28]. Inhibition of NETs formation significantly alleviates disease progression and improves kidney functions [27–29]. Adenine nephropathy is a type of crystalline nephropathy that leads to severe tubulointerstitial nephritis, renal scarring, and finally, end-stage renal disease. Excessive intake of adenine is associated with its precipitation within the kidneys, leading to damage of tubular cells, necrosis, and subsequent infiltration of immune cells [30]. So far, no experimental study has been published focusing on the role of NETs formation in adenine-induced nephropathy. Therefore, we aimed to describe the role of NETs formation in a mouse model of adenine-induced CKD using wild-type mice as well as Padi4$^{-/-}$ mice. Padi4$^{-/-}$ mice are widely used to elucidate the involvement of NETs in various pathologies [9,31,32]

and should, thus, be suitable as a genetic tool to study the role of NETs in adenine-induced nephropathy. We hypothesized that Padi4$^{-/-}$ mice will have lower NET-associated markers than their wild-type counterparts and that PAD4 deficiency would be protective against the development of the induced CKD. The study of pathomechanisms involved in the development of CKD in animal models may hold clinical relevance. Targeting molecules that are implicated in NETs formation or degradation is a feasible tool and has already been tested clinically in other diseases. Thus, this approach could lead to novel potential therapeutic strategies for CKD.

## Materials and methods

This study was conducted according to ARRIVE guidelines for the studies performed on laboratory animals and was approved by the Ethics Committee of the Institute of Pathophysiology, Faculty of Medicine, Comenius University, Bratislava, Slovakia, and the State Veterinary and Food Administration of the Slovak Republic (2948/2023).

### Animals and housing conditions

In this study, young male C57BL/6J mice (wild-type mice – WT, n = 17, 10–12 weeks old, Velaz, Prague, Czech Republic) and Padi4$^{-/-}$ mice (B6. Cg-Padi4tm1.1Kmow/J, n = 15, 10–12 weeks old, Jackson Laboratory, Bar Harbor, Maine, United States, JAX stock #030315) with no prior interventions were used. The Padi4$^{-/-}$ males were generated by mating pairs of homozygous Padi4$^{-/-}$ males and females in the institutional animal facility. Due to small litter sizes, animals used in this experiment originated from several litters and were age-matched. All animals were group-housed (4–5 per cage) in individually ventilated cages (Tecniplast, 1285L, 36.5 x 20.7 x 13 cm), in a room with a controlled light/dark cycle (12/12 hours), constant temperature (22±2°C), and humidity (45–65%). All animals had *ad libitum* access to standard rodent diet (standard diet for mice, Eypy-KMK20, Bruzovice, Czech Republic) and water.

### Design of the experiment

Mice of both genotypes were randomly divided according to body weight into 2 groups: adenine and control groups, with a single animal being an experimental unit. Adenine nephropathy was induced by daily intraperitoneal application of adenine hydrochloride hydrate (100 mg/kg, Sigma Aldrich, Munich, Germany) dissolved in saline to WT mice (n = 10) and Padi4$^{-/-}$ mice (n = 9) for 14 consecutive days. To minimize pain and discomfort, the side of the injection was alternated each day. Age-matched WT mice (n = 7) and Padi4$^{-/-}$ mice (n = 6) in the control groups received daily intraperitoneal injections of saline for 14 days. All animals within a cage underwent experimental procedures in the same order, and the sequence of measurements between cages was alternated to reduce order bias. Experimenters conducting metabolic and physiological assessments were blinded to genotype and treatment during data collection. All animals within a cage underwent experimental procedures in the same order, and the sequence of measurements between cages was alternated to reduce order bias. Experimenters conducting metabolic and physiological assessments were blinded to genotype and treatment during data collection.

### Sample collection

For the urine collection, mice were placed into individual metabolic cages for 3 hours. Urine was collected at baseline, after 7 and 14 days into sterile Eppendorf tubes. Venous blood was collected at the same time points (baseline, after 7 and 14 days) from the retroorbital plexus into K$_3$EDTA and lithium-heparin tubes (Sarstedt, Nümbrecht, Germany) after previous inhalation anesthesia using Isoflurane (3%, Vetpharma Animal Health, Barcelona, Spain) mixed with 97% oxygen. To reduce invasiveness, venous blood for the assessment of renal parameters after 3 and 10 days was collected from the tail vein of mice without the need of inhalation anesthesia. Following terminal blood collection after 14 days, animals were euthanized by cervical dislocation after previous inhalation anesthesia (3% Isoflurane, 97% oxygen). Likewise,

if an animal presented with either five or more mild signs of suffering (e.g., < 20% weight loss, markedly reduced food or water intake, dehydration, reduced skin elasticity, ocular abnormalities, postural changes, or atypical urine or feces) or with two severe signs of distress (e.g., > 20% weight loss, persistent ocular discharge, continuous tremors or vocalization, repetitive or self-injurious behavior, lack of responsiveness, or sustained rigidity), it was euthanized. Immediately after collection, blood and urine samples were centrifuged at 1600 x g, 4°C for 10 minutes, and supernatants were stored at −20°C for subsequent assessment of renal functions and NET-associated markers.

## Biochemical analysis

Plasma creatinine and blood urea nitrogen (BUN) concentrations were assessed using commercially available spectro-photometric assays following the manufacturer's instructions (Creatinine Serum Low Sample Volume Kit; Urea Nitrogen Colorimetric Detection Kit; Arbor Assays, Ann Arbor, USA). For plasma creatinine measurement, 15 µl of plasma was mixed with an equal volume of Assay Diluent in a 384-well plate, after which 60 µl of DetectX Creatinine Reagent was added. Absorbance at 490 nm was measured at 1 and 30 minutes, and the final value was calculated by subtracting the 1-minute reading from the 30-minute reading. For BUN measurement, plasma was diluted 1:20 with distilled water, incubated with assay reagents for 30 minutes at room temperature, and absorbance was measured at 450 nm. Urinary creatinine was measured using the Jaffe method as described previously [33]. In brief, 10 µl of standards or samples were mixed with 200 µl of freshly prepared working solution (0.2 M NaOH and 25 mM picric acid in a 5:1 ratio), and absorbance was measured at 492 nm at both 1 and 6 minutes, with the initial reading serving as the blank.

Myeloperoxidase (MPO), neutrophil elastase (NE) and neutrophil gelatinase-associated lipocalin (NGAL) in plasma and urine were determined using commercially available ELISA kits (Mouse Myeloperoxidase DuoSet Elisa Kit, Mouse Neutrophil Elastase/ELA2, Mouse Lipocalin-2/NGAL DuoSet Elisa Kit, R&D Systems, Minneapolis, USA) as recommended by the manufacturer. For MPO and NE measurements, plasma samples were 100-times diluted in Reagent Diluent (1% BSA in PBS), while urine samples were 10-times diluted. For NGAL, plasma was diluted 10000-times and urine was diluted 100000-times in the same manner. Absorbance was measured at 450 nm. Nucleosomes in plasma were quantified using commercial photometric assay (Cell death detection ELISA kit, Roche, Basel, Switzerland) and are expressed in arbitrary units (AU).

## Extracellular DNA isolation

Plasma and urine samples for ecDNA isolation were prepared by sequential centrifugation of samples at 1600 × g for 10 minutes followed by 16,000 × g for 10 minutes, both at 4 °C. Two hundred ul of the resulting supernatants were then processed using a commercial kit (QIAamp DNA Mini Kit; Qiagen, Hilden, Germany) following the manufacturer's instructions. Fifty ul of ultrapure water was used for the elution of ecDNA.

## Extracellular DNA quantification

Total ecDNA in plasma and urine was quantified using a Qubit Fluorometer and Qubit dsDNA HS Assay Kit (Invitrogen, Carlsbad, CA, USA). Subcellular origin of ecDNA was determined using real-time PCR. Nuclear DNA (ncDNA) was quantified using primers targeting the mouse beta-2-microglobulin gene (F: 5´-TGT CAG ATA TGT CCT TCA GCA AGG-3´, R: 5´-TGC TTA ACT CTG CAG GCG TAT G-3´), while primers for mitochondrial DNA (mtDNA) were designed to amplify mouse cytochrome c (F: 5´-CCC AGC TAC TAC CAT CAT TCA AGT-3´, R: 5´-GAT GGT TTG GGA GAT TGG TTG ATG T-3´). Each PCR reaction (final volume of 10 µl) contained 5 µl of SsoAdvanced Universal SYBR® Green Supermix (2×) (Bio-Rad, Hercules, CA, USA), 0.25 µL of each 10 µM primer (Microsynth, Balgach, Switzerland), 2.5 µL of template, and 2 µL of nuclease-free water. Real-time PCR program included an initial denaturation at 98 °C for 3 minutes, followed by 40 cycles of denaturation at 98 °C for 15 seconds, annealing at 51 °C for 30 seconds for ncDNA and 47 °C for 30 seconds

for mtDNA, respectively, and extension at 60 °C for 30 seconds. After 40 cycles, melting curve was performed to prove the specificity. The concentration of ncDNA and mtDNA in urine is expressed in genomic equivalent (GE) per milligram of creatinine.

### Histological analysis

Picro Mallory Trichrome staining was performed to visualize fibrosis, with a blue stain of collagen. Kidneys were cut into 4 µm paraffin sections, dewaxed and processed according to the manufacturer's protocol. The representative images were cached at a magnification of × 400. Immunohistochemistry for Ly-6G (Cell Signaling, 87048), MPO (Abcam, ab208670), citH3 (Abcam, ab52946), and PAD4 (Abcam, ab214810) was performed on kidneys taken at the end of the experiment. Kidneys were cut into 3 µm paraffin sections, dewaxed and subjected to antigen retrieval by 15 minutes incubation in 10 mM citrate buffer, pH 6.0, at 80 °C (Ly6G and citH3) or 10 mM TRIS/EDTA pH 9 (MPO and PAD4). The concentrations of antibodies were 1:100, 1:100, 1:200, and 1:200 in 1%BSA/PBS, respectively. For all proteins, a 2-step immunoperoxidase technique was used according to standard techniques. Peroxidase activity was developed using diaminobenzidine and $H_2O_2$. The representative images were cached at a magnification of × 100 and × 400. Ten fields per one kidney slice were evaluated by an independent scientist (cell count for Ly6G and citH3, positive areas for MPO, PAD4).

### Statistical analysis

GraphPad Prism 6.01 (GraphPad Software Inc, La Jolla, USA) was used for statistical analysis. To compare concentrations of ecDNA and neutrophil granular proteins including MPO, NE and NGAL between groups, three-way ANOVA (one factor was genotype, the second was treatment, and the third was time) and subsequently Tukey post hoc test were used. To compare the concentration of plasma nucleosomes after 14 days, two-way ANOVA (one factor was genotype, the second was treatment) with Bonferroni-corrected t-test test was used for the post hoc analysis. Outliers were detected using Grubbs test. All data are expressed as mean ± SD. A $p$-value less than 0.05 was considered statistically significant.

## Results

### Assessment of body weight and kidney functions in mice

To prevent excessive body weight loss, animals were weighed daily (Fig 1A). Three-way ANOVA showed a significant effect of adenine injection (F = 72.72, p < 0.001) and its duration (F = 49.39, p < 0.001), while genotype did not affect the body weight of animals (F = 2,890, p > 0.05). Padi4$^{-/-}$ mice treated with adenine exhibited a reduction of body weight by 13% already 2 days after adenine treatment (q = 5.87, p < 0.05), while WT mice had a significant decrease in body weight after 4 days (by 15%, q = 7.14, p < 0.01) when compared to corresponding healthy controls.

To verify CKD induction, plasma creatinine and BUN were assessed (Fig 1B and 1C). Three-way ANOVA revealed a significant effect of adenine administration (F = 178.1, p < 0.001), genotype (F = 11.59, p < 0.01), and treatment duration (F = 28.63, p < 0.001) on plasma creatinine. In Padi4$^{-/-}$ mice that received adenine, plasma creatinine was higher by 150% after 3 days (q = 11.39; p < 0.001) in comparison to the control group. Plasma creatinine in WT adenine-treated mice was twice as high as in the control group after 7 days (q = 7.25; p < 0.001). When comparing WT and Padi4$^{-/-}$ mice receiving adenine, there was a significant difference between the two groups on day 3 (q = 10.95; p < 0.001) and day 7 (q = 5.85; p < 0.01), where Padi4$^{-/-}$ adenine-treated mice showed higher plasma creatinine concentrations compared to WT adenine-treated mice. This difference disappeared on day 10 (q = 0.77; p > 0.05). Three-way ANOVA showed a significantly effect of adenine injection (F = 140.7, p < 0.001) and treatment duration (F = 32.43, p < 0.001), but not genotype (F = 0.31, p > 0.05) on plasma BUN concentrations. Plasma BUN in WT and Padi4$^{-/-}$ adenine treated mice was 3-times and 4-times higher after 7 days (WT: q = 17.41, p < 0.001; Padi4$^{-/-}$: q = 21.74, p < 0.01) compared to corresponding control groups and remained elevated after 14 days (WT: q = 8.15, p < 0.01; Padi4$^{-/-}$: q = 10.55, p < 0.01). However, no significant differences

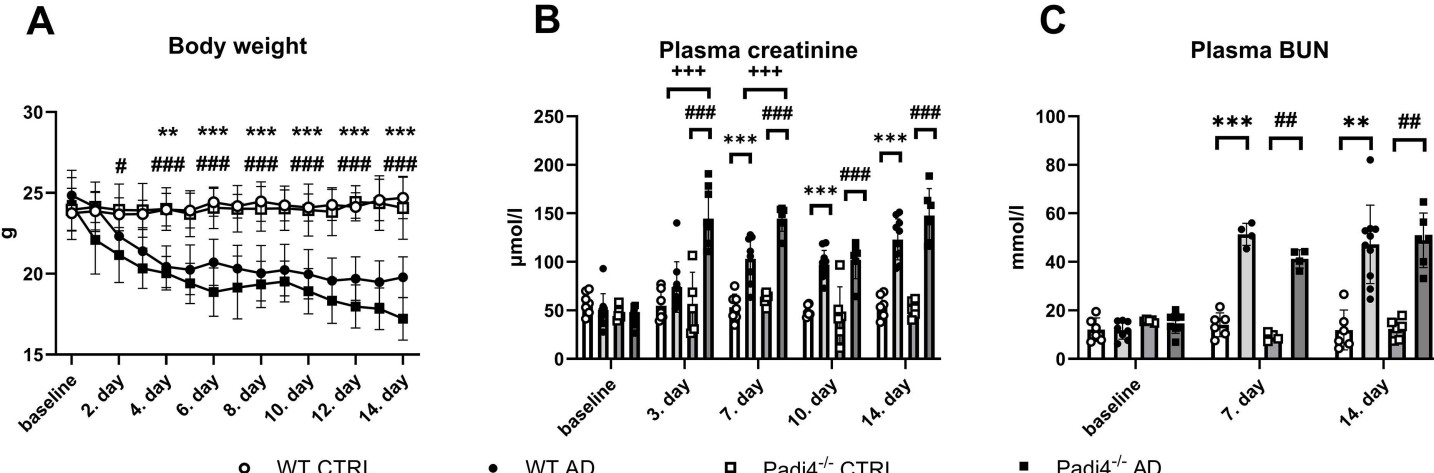

**Fig 1. Assessment of body weight changes and kidney functions in WT and Padi4$^{-/-}$ adenine-treated mice and corresponding control groups during 14 days. A:** Body weight change in WT and Padi4$^{-/-}$ mice receiving adenine or vehicle during the 14-day period. **B**: Concentrations of plasma creatinine in WT and Padi4$^{-/-}$ mice receiving adenine or corresponding control groups at baseline, and after 3, 7, and 14 days of treatment. **C:** Concentrations of plasma BUN in WT and Padi4$^{-/-}$ mice receiving adenine or vehicle at baseline, and after 7, and 14 days of treatment. Data were analyzed using three-way ANOVA and subsequently Tukey´s post hoc test. ** denotes $p < 0.01$ WT adenine-treated vs. corresponding control group, *** denotes $p < 0.001$ WT adenine-treated mice receiving adenine vs. corresponding control group, # denotes $p < 0.05$ Padi4$^{-/-}$ adenine-treated mice vs. corresponding control group, ## denotes $p < 0.01$ Padi4$^{-/-}$ adenine-treated mice vs. corresponding control group, ### denotes $p < 0.001$ Padi4$^{-/-}$ adenine-treated mice vs. corresponding control group, +++ denotes $p < 0.001$ WT adenine-treated mice vs. Padi4$^{-/-}$ adenine-treated mice. AD – adenine group, CTRL – control group receiving saline, Padi4$^{-/-}$– protein-arginine deiminase type 4 knockout mice, WT – wild-type mice.

were detected between WT and Padi4$^{-/-}$ adenine-treated groups at either 7 days (q = 4.86, p > 0.05) or 14 days (q = 0.35, p > 0.05).

## Measurement of NET-associated markers

**ecDNA, nucleosomes.** Plasma total ecDNA increased following adenine injection (F = 19.25, p < 0.001) and by the duration of the treatment (F = 12.51, p < 0.001), but was not affected by the genotype of mice (F = 0.44, p > 0.05, Fig 2A). Plasma ecDNA in both WT and Padi4$^{-/-}$ mice receiving adenine was higher by 88% and 250% after 7 days of adenine administration compared to corresponding control mice, however this rise was not significant (WT: q = 4.21, p > 0.05; Padi4$^{-/-}$: q = 3.16, p > 0.05). ecDNA did not differed between WT and Padi4$^{-/-}$ adenine-treated groups after 7 (q = 1.02, p > 0.05) and 14 days, respectively (q = 2.32, p > 0.05). Three-way ANOVA revealed a significant effect of adenine administration resulting in increased ncDNA (F = 8.34, p < 0.01) and mtDNA (F = 4.70, p < 0.05) in plasma. Similarly, the duration of adenine treatment showed a similar trend (ncDNA: F = 3.34, p = 0.07; mtDNA: F = 3.86, p = 0.06, Fig 2B and 2C). Genotype did not affect plasma concentrations of ncDNA (F = 0.008, p > 0.05) and mtDNA (F = 2.25, p > 0.05). Two-way ANOVA revealed a significant effect of adenine administration (F = 6.87, p < 0.01) on plasma nucleosomes, while genotype did not affect their plasma concentrations (F = 1.18, p > 0.05). After 14 days, WT mice with adenine nephropathy had significantly higher plasma nucleosomes compared to the control group (t = 2.57, p < 0.05). There was no significant difference between Padi4$^{-/-}$ adenine-treated mice and the corresponding control group (t = 1.22, p > 0.05, Fig 2D).

Three-way ANOVA showed that urinary total ecDNA increased following the adenine injection (F = 28.16, p < 0.001), by the duration of adenine treatment (F = 12.80, p < 0.01), and the genotype of mice (F = 7.59, p < 0.05, Fig 2E). Urinary ecDNA concentrations in WT and Padi4$^{-/-}$ mice receiving adenine were 4-times and 8-times higher after 7 days compared to the corresponding control groups, but it was not statistically significant (WT: q = 2.56, p > 0.05; Padi4$^{-/-}$: q = 4.45, p > 0.05). When comparing WT and Padi4$^{-/-}$ adenine-treated mice, ecDNA concentrations in the urine of WT adenine mice

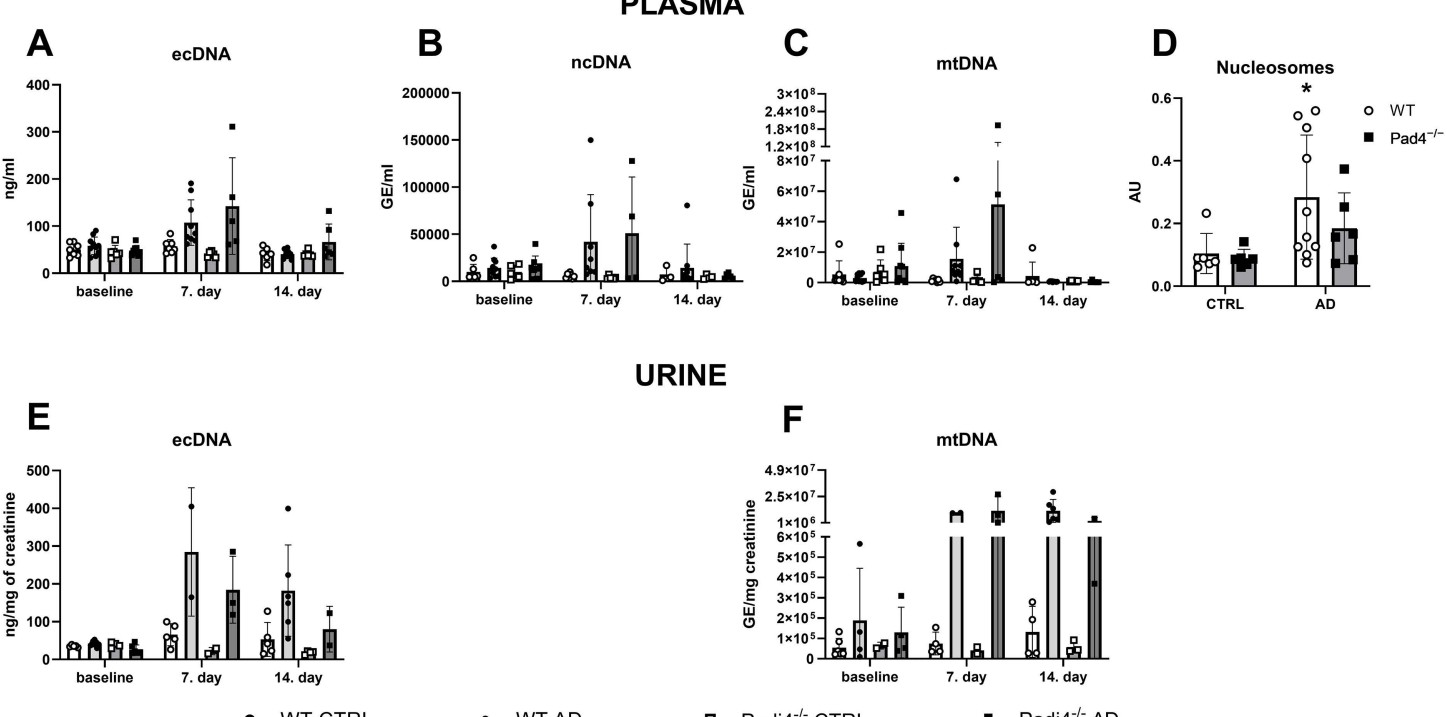

**Fig 2. Concentrations of ecDNA and nucleosomes in WT and Padi4⁻/⁻ adenine-treated mice and corresponding control groups.** Concentrations of **A**: total ecDNA, **B**: ncDNA, **C**: and mtDNA in plasma of WT and Padi4⁻/⁻ mice receiving adenine or vehicle at baseline, and after 7 and 14 days of treatment. **D**: Concentrations of plasma nucleosomes in WT and Padi4⁻/⁻ mice after 14 days of adenine or vehicle treatment. Concentrations of **E**: total ecDNA and **F**: mtDNA in urine of WT and Padi4⁻/⁻ mice at baseline, and after 7 and 14 days of adenine or control. Three-way ANOVA with Tukey's post hoc test was used for A–C, E–F; and two-way ANOVA with Bonferroni's test for **D**. Data are shown as individual values with mean ± SD. *denotes p < 0.05 in WT mice receiving adenine vs. the corresponding control group. No significant differences were found unless indicated. AD – adenine group, CTRL – control group receiving saline, ecDNA – extracellular DNA, mtDNA – mitochondrial DNA, ncDNA – nuclear DNA, Padi4⁻/⁻– protein-arginine deiminase type 4 knockout mice, WT – wild-type mice.

were higher on average by 50% (q = 1.09, p > 0.05) and 100% (q = 2.21, p > 0.05) compared to their Padi4⁻/⁻ counterparts after 7 days and 14 days of adenine treatment, respectively. Nevertheless, this difference was not statistically significant. Adenine treatment resulted in an increase in urinary mtDNA concentrations (F = 10.47, p < 0.01, Fig 2F), while the effect of the duration of adenine treatment (F = 3.29, p > 0.05) and the genotype of the mice (F = 0.51, p > 0.05) was not found. Urinary mtDNA concentrations in WT adenine-treated mice were significantly higher after 7 days compared to the control group (by 130-times, q = 78.78, p < 0.05). Similarly, Padi4⁻/⁻ adenine-treated mice had mtDNA concentration higher by 300-times after 7 days compared to the control group, however, this rise was statistically not significant (q = 2.18, p > 0.05). Padi4⁻/⁻ adenine-treated mice differ from WT adenine-treated mice in urinary mtDNA concentrations neither after 7 (q = 0.37, p > 0.05), nor after 14 days (q = 2.71, p > 0.05) of adenine administration.

**MPO, NE, NGAL.** Plasma MPO and NE were increased by adenine administration (MPO: F = 12.82, p < 0.01; NE: F = 60.45, p < 0.001) and by the duration of the treatment (MPO: F = 8.99, p < 0.001; NE: F = 20.31, p < 0.001; Fig 3A and 3B), while genotype did not affect their concentrations (MPO: F = 1.72, p > 0.05; NE: F = 0.21, p > 0.05). After 7 days of adenine administration, post hoc test revealed no significant difference in plasma MPO between WT or Padi4⁻/⁻ adenine-treated mice and corresponding control groups (WT: q = 4.44, p > 0.05; Padi4⁻/⁻: q = 2.45, p > 0.05). Similarly, there was no difference in plasma MPO between the 2 adenine groups (WT adenine-treated mice vs Padi4⁻/⁻ adenine-treated: q = 2.05, p > 0.05, Fig 3A). WT and Padi4⁻/⁻ groups receiving adenine had significantly higher plasma NE after 7 days (WT: by

**PLASMA**

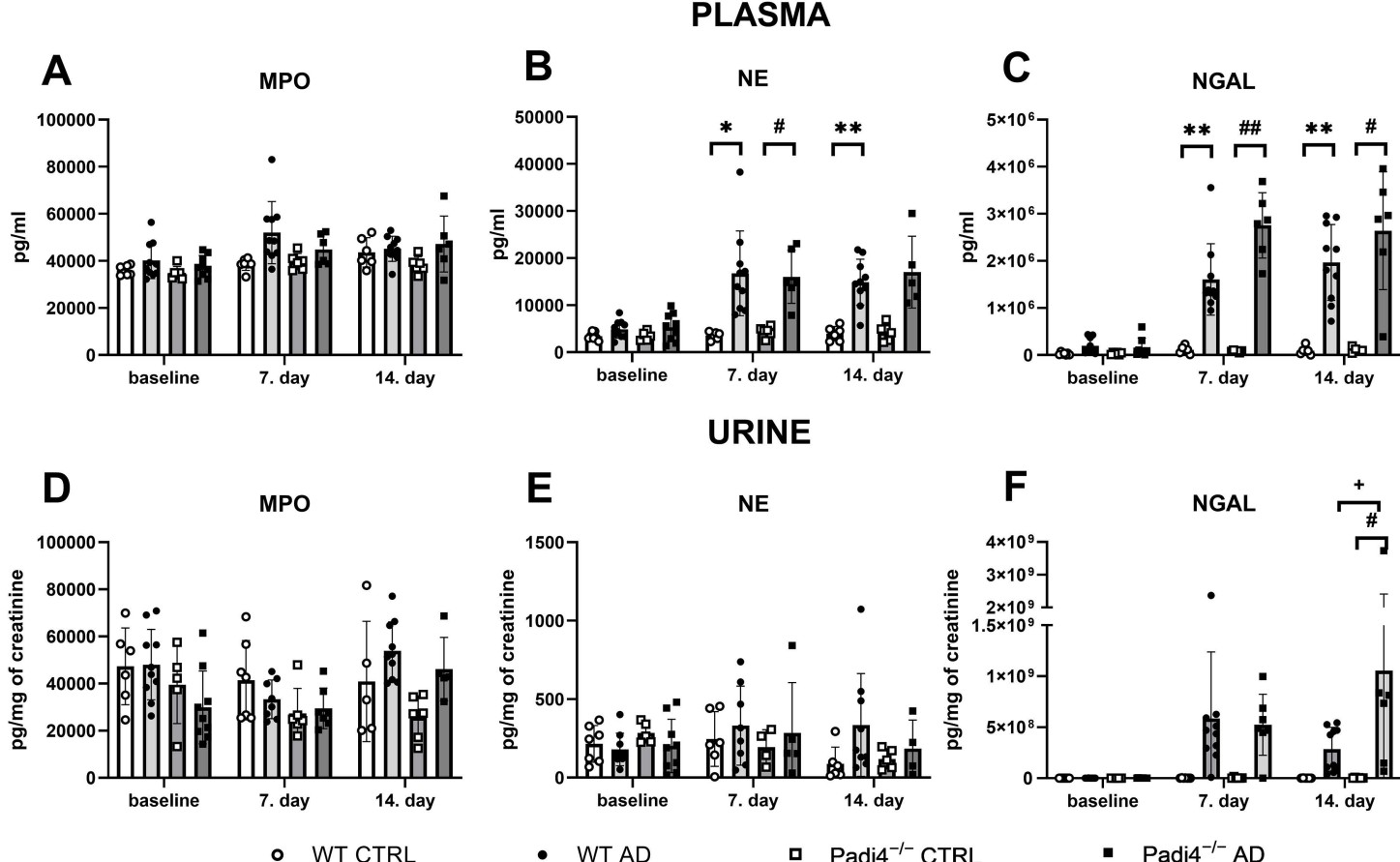

**Fig 3. Concentrations of neutrophil granular proteins in WT and Padi4⁻/⁻ adenine-treated mice and corresponding control groups.** Concentrations of **A**: MPO **B**: NE, **C**: NGAL in plasma of WT and Padi4⁻/⁻ mice receiving adenine or vehicle at baseline, and after 7 and 14 days of treatment administration. Concentrations of **D**: MPO, **E**: NE, **F**: NGAL in urine of WT and Padi4⁻/⁻ adenine-treated mice or corresponding control groups at baseline, and after 7 and 14 days of treatment. Data were analyzed using three-way ANOVA and subsequently Tukey's post hoc test. Results are expressed as individual plots with mean±SD. * denotes p<0.05 WT mice receiving adenine vs. corresponding control group, ** denotes p<0.01 WT mice receiving adenine vs. corresponding control group, # denotes p<0.05 Padi4⁻/⁻ adenine-treated mice vs. corresponding control group, ## denotes p<0.01 Padi4⁻/⁻ adenine-treated mice vs. corresponding control group, + denotes p<0.05 WT adenine-treated mice vs. Padi4⁻/⁻ adenine-treated mice. AD – adenine group, CTRL – control group receiving saline, MPO – myeloperoxidase, NE – neutrophil elastase, NGAL – neutrophil gelatinase-associated lipocalin, Padi4⁻/⁻– protein-arginine deiminase type 4 knockout mice, WT – wild-type mice.

370%, q=6.53, p<0.05; Padi4⁻/⁻: by 264%, q=7.00, p<0.05) and 14 days (WT: by 275%, q=9.29, p<0.01; Padi4⁻/⁻: by 298%, q=5.18, p>0.05) of treatment administration in comparison to control mice (Fig 3B). Two adenine groups (WT adenine-treated vs Padi4⁻/⁻ adenine-treated mice) did not differ in the concentrations of plasma NE after 7 days (q=0.29, p>0.05) and 14 days (q=0.81, p>0.05, Fig 3B).

Plasma NGAL was increased by adenine injection (F=147.2, p<0.001), duration of treatment (F=35.65, p<0.001), and the genotype of mice (F=5.93, p<0.05, Fig 3C). Both adenine-treated groups had significantly higher plasma NGAL after 7 days (WT: 14-times, q=8.74, p<0.01; Padi4⁻/⁻: 33-times, q=13.34, p<0.01) and after 14 days (WT: 21-times, q=10.28, p<0.01; Padi4⁻/⁻: 24-times, q=7.00, p=0.06) than corresponding control groups. Padi4⁻/⁻ adenine-treated mice did not differ from WT adenine-treated mice plasma NGAL concentrations either after 7 (q=4.38, p>0.05), or 14 days (q=1.67, p>0.05) of adenine administration (Fig 3C).

Neither urinary MPO (F = 0.79, p > 0.05) nor NE (F = 1.74, p > 0.05) was affected by the adenine injection (Fig 3D and 3E). Urinary MPO was affected by the duration of adenine administration (F = 3.41, p < 0.05) and by the genotype (F = 11.26, p < 0.01), where after 14 days of treatment WT mice receiving adenine had by 20% higher urinary MPO than Padi4$^{-/-}$ adenine-treated mice, but this difference was statistically not significant (q = 1.52, p > 0.05, Fig 3D). However, the duration of the treatment (F = 0.97, p > 0.05) and genotype did not affect urinary NE concentrations (F = 0.20, p > 0.05, Fig 3E). After 14 days of adenine administration, WT and Padi4$^{-/-}$ mice had urinary NE concentrations higher by 280% and 55%, respectively, compared to corresponding control mice (WT: q = 2.97, p > 0.05; Padi4$^{-/-}$: q = 0.99, p > 0.05, Fig 3E). Moreover, urinary NE in WT adenine-treated mice was higher by 80% after 14 days of treatment administration compared to Padi4$^{-/-}$ adenine-treated mice (WT: q = 1.50, p > 0.05). However, these differences were not statistically significant.

Adenine treatment (F = 19.25, p < 0.001) and its duration (F = 5.00, p < 0.05, Fig 3F) led to an increase in urinary NGAL, but the genotype of mice (F = 1.65, p > 0.05) had no effect on its concentration. Urinary NGAL in WT adenine-treated mice was higher by 400-times and 280-times after 7 and 14 days of adenine injection compared to WT control mice. However, this increase was not significant (q = 3.93, p > 0.05; q = 1.83, p > 0.05). Padi4$^{-/-}$ mice receiving adenine had significantly higher urinary NGAL concentrations compared to Padi4$^{-/-}$ control mice following 14 days of treatment application (q = 6.05, p < 0.01). In addition, Padi4$^{-/-}$ adenine-treated mice showed significantly higher urinary NGAL concentrations after 14 days of treatment in comparison to WT adenine-treated mice (q = 4.96, p < 0.05, Fig 3F).

**Correlation between NET-associated markers.** Plasma ecDNA positively correlated with plasma MPO, NE, and NGAL (S1 Table). In addition, after 14 days of treatment application, concentrations of plasma ecDNA were positively associated with levels of plasma nucleosomes (r = 0.42, p < 0.05). Similarly, there were significant positive correlations between urinary ecDNA and MPO, NE, or NGAL. Moreover, plasma ecDNA and NGAL positively correlated with their urinary concentrations (S1 Table).

## Discussion

To the best of our knowledge, this is the first study describing the role of NETs formation in adenine-induced CKD, an animal model of crystalline nephropathy using WT and Padi4$^{-/-}$ mice. Our data indicate that circulating concentrations of NE, NGAL, ecDNA, and nucleosomes, as main structural components of NETs [34], are elevated in this mouse model of CKD, with a trend of increase also observed in urine. Moreover, we hypothesized that the absence of PAD4, an enzyme crucial for chromatin decondensation and subsequent NETs extrusion, would be associated with lower levels of NETs-associated markers, and hence this would confer protection against severe kidney injury in Padi4$^{-/-}$ adenine-treated mice. However, this was not confirmed since we have found no differences in plasma and urinary concentrations of NET-associated markers between WT and Padi4$^{-/-}$ mice. Additionally, prominent fibrosis together with rapidly decreased kidney functions were observed in both adenine-treated groups, regardless of the genotype.

In CKD, uremic conditions can excessively activate neutrophils, causing a pro-inflammatory environment and NETs formation [35]. However, studies show conflicting results, with some indicating higher circulating NETs markers in hemodialysis patients, correlating with endothelial dysfunction [22], while others suggest a reduced NETs-forming ability, increasing infection susceptibility [36]. Excessive NETs formation is implicated in the pathophysiology of specific CKD forms, such as lupus nephritis, where impaired clearance of NETs exacerbates exposure to autoantigens [10,37,38], causing vascular injury and inflammation [18,39]. NETs correlate with disease activity in lupus [40,41], and their suppression improves inflammation and kidney function [42,43]. NETs formation also contributes to other autoimmune disorders affecting the kidneys, including IgA or ANCA-associated vasculitis, with elevated NETs markers correlating with kidney lesions [44–46]. Further, in a mouse model of heterologous anti-GBM nephritis, NETs-related histones have a direct cytotoxic effect on glomeruli. Injection of PAD4 inhibitor or neutralizing anti-histone therapies reduces glomerular damage and decreases proteinuria [11]. Similarly, in an anti-GBM nephritis model, eliminating NETs via deoxyribonuclease 1 attenuates hematuria and glomerular dysfunction [12]. Importantly, a recent study identified the presence of NETs in the kidneys of

adenine-treated mice, and inhibition of NETs formation ameliorated kidney damage [47]. These results are in contrast to ours, since PAD4 deficiency in adenine-treated mice did not prevent kidney failure. However, important methodological differences between our findings and the previously published study should be considered. In the study by Jones et al. [47], NETosis was reduced through activation of the farnesoid X receptor, which suppresses sphingosine-1-phosphate signaling, thereby modulating renal inflammation and neutrophil activation. In contrast, our study used a genetic model in which PAD4 is absent in all tissues throughout life. Compensatory mechanisms via other PAD enzymes or PAD-independent pathways could explain the difference from pharmacologic NET-targeting interventions.

PAD4 has been seen as the key enzyme responsible for driving NETs formation by citrullinating histones. However, recent research suggests, that depending on the type of stimuli, other PAD enzymes, particularly PAD2, might also play a crucial role [48]. In mouse models of sepsis, PAD4 inhibition did not affect survival, while PAD2 or both PAD2 and PAD4 inhibition significantly reduced inflammation and improved survival in mice [49,50]. The study by Kenny et al. indicates that stimulating neutrophils with phorbol 12-myristate 13-acetate induces NETs release without histone citrullination, suggesting that at least some stimuli can trigger NETs formation without this modification [51]. Moreover, a recent study revealed that hyperuricemia induced by adenine administration can activate a non-canonical NETosis pathway dependent on Gasdermin D and caspase-11 [52], with similar findings reported in folic acid [53] and obstructive nephropathy [54]. This pathway occurs independently of the classical PAD-mediated axis [55] and may partly explain the differences between pharmacological PAD inhibition reported in previous studies and the observations obtained in our experimental model. It could be that in this model, or at least in Padi4$^{-/-}$ mice receiving adenine, NETs could be formed via this alternative pathway.

Beyond CKD, numerous studies confirmed the role of NETs in trapping crystals of various origins, playing an important role in sterile inflammation [26,27,56]. In acute gouty arthritis, monosodium urate crystals trigger pro-inflammatory cytokine production, recruiting neutrophils and releasing NETs, which contribute to acute gouty pain attacks. On the other hand, aggregated NETs components degrade inflammatory factors, resolving inflammation during the chronic phase of gout [56,57]. Moreover, studies show that suppression of crystal-induced NETs formation via various pathways ameliorates sterile inflammation in gout [58–60]. Further, evidence from the studies indicates that NETs play a crucial role in the progression of oxalate-induced crystalline nephropathy. Elevated levels of NET markers – including ecDNA, MPO-DNA, NE, and citrullinated H3 histone (citH3) correlate with tubular injury. However, suppression of NETs leads to a significant reduction of oxalate deposits and tissue damage, leading to improved kidney function [27,28]. In our research, we assumed that the precipitation of adenine would result in the recruitment of neutrophils to the kidneys, their activation, and subsequent NETs formation. Our results confirm higher concentrations of main NET-related components, including ecDNA, MPO, and NE, in the plasma and urine of mice receiving adenine compared to control mice. Additionally, a recent study of Ma et al. (2025) has shown that protein-bound DNA fraction detected in urine originates predominantly from the urinary tract rather than from transrenal sources [61], suggesting neutrophil infiltration into renal tissue – potentially followed by the release of NETs – or increased glomerular permeability to circulating NETs components occurring as a result of kidney damage. Therefore, we measured nucleosome levels in the plasma of mice to further investigate systemic NET-associated activity and its potential contribution to renal pathology. Increased concentrations of nucleosomes were found in adenine-treated mice regardless of the genotype. Nevertheless, due to low urine volume, we did not evaluate their concentrations in urine, which remains to be investigated in future studies.

However, the hypothesis about lower NETs formation in Padi4$^{-/-}$ mice that would protect mice from severe kidney damage was not supported by our data. We observed no differences in plasma and urinary NET-associated markers between WT and Padi4$^{-/-}$ adenine-treated mice, and correspondingly, no differences in renal functions, histological findings, and immunohistochemical analyses between these two genotypes (S1 and S2 Figs). Importantly, the markers assessed in this study (MPO, NE, ecDNA, and nucleosomes) are not specific for NETs and may also reflect neutrophil activation independent of NETs formation or release from other cell types.

Although neutrophils are considered the primary source of extracellular traps, other immune cells, including macrophages, eosinophils, and basophils, have also been shown to release extracellular traps in response to crystalline stimuli

such as monosodium urate or calcium phosphate crystals [62–64]. The precise mechanisms of NETs formation are still not fully elucidated, and even less is known about the formation of ETs via other immune cells [26]. Moreover, the type of crystal-induced cell death may depend on the size and composition of the crystals, with larger crystals predominantly inducing NETs formation [63,65]. Nevertheless, individual NET-associated markers such as MPO, NE, or ecDNA might originate from other inflammatory processes and do not have to be specific for NETs [66].

Currently, for the detection and quantification of NETs formation, several methods are available, but none of them is considered a gold standard. Co-localization of at least two markers is generally accepted as a NET. Accordingly, the assessment of DNA-MPO or DNA-citH3 complexes using ELISA is a widely accepted approach for NETs quantification [67]. However, in our conditions, this method did not provide the quantitative accuracy needed for the analysis. As a result, we opted to use kits that detect individual NET-associated components rather than these complexes. Nevertheless, to enhance the reliability of NETs detection, we decided to measure more than two components. Another limitation of our study is the lack of direct NETs detection in kidney tissue. Instead, citH3 was evaluated as a marker associated with NETs formation, reflecting chromatin decondensation during NETosis (S1 Fig). Nevertheless, more precise methods, such as immunofluorescence microscopy with co-localization of multiple markers, should be employed in future studies. In addition, only male mice were included, and thus potential sex-related differences in the response to adenine-induced CKD and PAD4 deficiency were not assessed. Although Padi4 deficiency does not appear to affect NETs formation and severity of renal failure in a mouse model of adenine-induced CKD, alternative strategies to inhibit NETs production or enhance their clearance could represent potential therapeutic approaches, which will be investigated in future studies.

## Conclusions

This is the first study to investigate the role of NETs formation in adenine-induced nephropathy using WT and Padi4$^{-/-}$ mice. Increased levels of NET-associated markers were observed in both genotypes, suggesting that neutrophil activation and NETs formation occur during the progression of adenine-induced kidney injury. However, Padi4 deficiency did not protect mice from kidney injury, indicating that PAD4-dependent NETosis is unlikely to represent the principal mechanism driving disease progression in this model. These findings suggest that alternative pathways of NETs formation, including compensation by other PAD enzymes or PAD-independent NETs formation mechanisms, may be involved. Further experiments using pharmacological PAD inhibition in WT animals are required to provide additional mechanistic insight and clarify the contribution of alternative NETs formation pathways in this model.

## Supporting information

**S1 Fig. Representative images of Picro Mallory Trichrome fibrosis staining in WT and Padi4$^{-/-}$ mice with adenine nephropathy or corresponding control groups.** Picro Mallory Trichrome staining was used to assess renal fibrosis visualized by blue colour as well as tubular dilatation. An abundant interstitial fibrosis and tubular dilatation was found in both adenine-treated groups (WT and Padi4$^{-/-}$; black arrow and dashed arrows, respectively), whereas no fibrosis and no tubular dilatation was found in both control saline-treated groups (WT and Padi4$^{-/-}$). Images were cached at a magnification of 400 x.
(TIF)

**S2 Fig. Representative images of Ly6G, MPO, citH3, and PAD4 expression in WT and Padi4$^{-/-}$ mice with adenine nephropathy or corresponding control groups.** Ly6G, MPO, citH3, and PAD4 immunostaining were performed on sections from all four groups of mice. Ly6G expression was found in both genotypes treated with adenine, whereas no expression was found in both control groups. Moreover, abundant MPO expression was found in both, interstitium and glomeruli in adenine-treated groups of both genotypes, whereas no expression was found in both control groups. Additionally, citH3 expression was found in tubuli of both genotypes treated with adenine, whereas no or little expression was

found in both control groups. PAD4 staining was found in adenine-treated WT, whereas no positive staining was found in Padi4 $^{-/-}$ mice with and without adenine treatment, as well as in WT mice without adenine treatment. Images were cached at a magnification of × 100 and × 400. Dark brown spots revealed positive staining. AD – adenine-treated group, CTRL – control group receiving saline, citH3 – citrullinated histone H3, Ly6G – lymphocyte antigen 6 complex locus G, MPO – myeloperoxidase, Padi4$^{-/-}$– protein-arginine deiminase type 4 knockout mice, WT – wild-type mice.
(TIF)

**S1 Table. Relation between NET-associated markers in plasma and in urine of all animals, regardless of genotype or treatment. ecDNA – extracellular DNA, MPO – myeloperoxidase, NE – neutrophil elastase, NGAL – neutrophil gelatinase-associated lipocalin, r – Spearman´s rank-order correlation coefficient.** Values $p \leq 0.05$ are considered statistically significant.
(DOCX)

**S1 Methods. Histological and immunohistochemical analysis of kidney sections, including fibrosis assessment and detection of Ly6G, MPO, citH3, and PAD4.**
(DOCX)

## Author contributions

**Conceptualization:** Alexandra Gaál Kovalčíková, Veronika Borbélyová, Ľudmila Podracká, Peter Celec.

**Data curation:** Alexandra Gaál Kovalčíková, Gabriela Forraiová, Nelia Korotushak, Veronika Borbélyová, Diana Vavrincová-Yaghi, Peter Vavrinec.

**Formal analysis:** Alexandra Gaál Kovalčíková, Veronika Borbélyová.

**Funding acquisition:** Alexandra Gaál Kovalčíková.

**Investigation:** Alexandra Gaál Kovalčíková, Diana Vavrincová-Yaghi, Peter Vavrinec.

**Methodology:** Alexandra Gaál Kovalčíková, Veronika Borbélyová, Ľudmila Podracká, Peter Celec.

**Project administration:** Peter Celec.

**Resources:** Ľudmila Podracká.

**Supervision:** Ľudmila Podracká, Peter Celec.

**Writing – original draft:** Alexandra Gaál Kovalčíková.

**Writing – review & editing:** Ľudmila Podracká, Peter Celec.

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
