## [Decision Letter · Decision Letter 0]

11 Feb 2026

PONE-D-25-62916Neutrophil extracellular traps in the animal model of adenine-induced chronic kidney diseasePLOS One

Dear Dr. Celec,

Thank you for submitting your manuscript to PLOS ONE. First, I would like to apologize for the delay in the reviewing due to difficulties to find reviewers. After careful consideration, we feel that your manuscript has merit but does not fully meet PLOS ONE’s publication criteria as it currently stands. Therefore, we invite you to submit a revised version of the manuscript that addresses the points raised during the review process. Please, provide in particular more details about the origin and housing of mice. The demonstration of netosis in the tissue is mandatory. Compensatory mechanisms should also be further investigated, knowing the discrepancy between the present study and a previously published one using Pad inhibitor. Generally speaking, the manuscript conclusions are somehow premature.

We look forward to receiving your revised manuscript.

Kind regards,

Michel Simon, Ph. D.

Academic Editor

PLOS One

Journal Requirements:

“This work was supported by grants from the Grant Agency of Ministry of Education, Science, Research and Sport of the Slovak Republic No. VEGA 1/0212/22 and 1/0801/25 (AGK).”

“This work was supported by grants from the Grant Agency of Ministry of Education, Science, Research and Sport of the Slovak Republic No. VEGA 1/0212/22 and 1/0801/25 (AGK).”

4. We notice that your supplementary figures/ tables are included in the manuscript file. Please remove them and upload them with the file type 'Supporting Information'. Please ensure that each Supporting Information file has a legend listed in the manuscript after the references list.

Additional Editor Comments (if provided):

In the study reported in this manuscript, Peter Celec and his team investigated whether neutrophil extracellular traps (NETs) are involved in mouse adenine-induced nephropathy, as a model of chronic kidney diseases. To test this hypothesis, they compared wildtype and Padi4-/- adult mice. They analyzed the renal function and NET-related biomarkers in the plasma and urine of animals during the disease induction. No significant differences were observed between the Padi4-/- and wild-type mice, except a slightly earlier induction of the disease in genetically deficient animals (higher plasma creatinine level and related-body weight in the first days following adenine injections). The authors concluded that NETs are not directly involved in the pathogenesis.

Although this study is of interest, the conclusion seems to be premature.

The main point concerns the presence of NETs in the kidney tissue that was not directly assessed. The authors made the hypothesis that the amounts of NET markers (MPO, NE, extracellular DNA) in the urine/plasma is representative of NETs in the tissue. They must visualized and quantified netosis in the kidney, using citrullinated-histone antibodies. The infiltration of immune cells, including neutrophils, as well.

The authors suggested that other Pads may compensate the absence of Pad4, this has to be tested. In particular, Pad inhibitors has to be used in wildtype animals.

In a previously published study, netosis was reported to be associated with adenine-induced nephropathy in mice [47]. This study was consistent with the many different studies showing a role of netosis in the human chronic kidney diseases and in other mouse models of the disease; as mentioned by the authors themselves. How do the authors explain this discrepancy? They have to test their hypotheses (see the previous point).

Minor points

What is the genetic background of Padi4-/- mice?

What is the response of female mice?

Please give scale bars on (immuno)histological pictures, rather than magnifications.

Please give the origin of antibodies.

The rise in plasma ecDNA was higher in Padi4-/- as compared to wild-type mice, but the difference was not statistically significant. Did the authors consider to increase the number of analyzed mice? For other non-statistically significant different values as well.

Reviewers' comments:

Reviewer's Responses to Questions

**Comments to the Author**

1. Is the manuscript technically sound, and do the data support the conclusions?

Reviewer #1: Partly

2. Has the statistical analysis been performed appropriately and rigorously? 

Reviewer #1: Yes

3. Have the authors made all data underlying the findings in their manuscript fully available?

Reviewer #1: Yes

4. Is the manuscript presented in an intelligible fashion and written in standard English?

Reviewer #1: Yes

5. Review Comments to the Author

Reviewer #1: Kovalčíková et al. present a manuscript where they use PAD4-deficient animals to investigate how NETs can potentially contribute to chronic kidney disease in a model of adenine-induced nephropathy. They find that WT and PAD4-/- mice have comparable responses, except an earlier exacerbated response in PAD4-/- animals.

Could the authors provide more information about the origins and colony management of the mice? Housing conditions can impact immune responses and it is not clear if these are littermates or kept as separate colonies either at the institution or from the supplier.

The authors note in their discussion they measured NET components rather than NETs themselves. They hypothesize that NETs can be released independent of PAD4. Another plausible explanation is that the markers measured do not come from NETs but rather come from other cells (in the case of DNA or nucleosomes) or from neutrophil activation independent of NETosis (i.e. degranulation releasing MPO or NE). The authors should either provide data with more specific NET measurements, such as H3Cit-DNA or MPO-DNA, or dramatically reword the text to avoid conclusions about NETs altogether but rather describe the components measured. From the data provided it is not clear whether NETs are even being formed in this model.

As a minor comment, the authors should either use the gene name (Padi4-/-) or the protein abbreviation (PAD4-/-) rather than Pad4-/- through the manuscript text.

6. PLOS authors have the option to publish the peer review history of their article (what does this mean?). If published, this will include your full peer review and any attached files.

Reviewer #1: No

---

## [Author Response · Author response to Decision Letter 1]

4 May 2026

Dear Editors,

Re: Manuscript submission PONE-D-25-62916

Please find attached the revised version of our manuscript entitled „Neutrophil extracellular traps in the animal model of adenine-induced chronic kidney disease.” We highly appreciate the efforts of the editor and reviewers who carefully reviewed the paper and offered valuable suggestions. We have tried to improve the manuscript according to the constructive comments of the reviewers. We hope that the corrected version of our manuscript is now suitable for publication in PLOS ONE. Here is our point-by- point response to each of the comments of the reviewers.

Editorial Board Member comment 1

Comment: Please, provide in particular more details about the origin and housing of mice. Compensatory mechanisms should also be further investigated, knowing the discrepancy between the present study and a previously published one using Pad inhibitor. Generally speaking, the manuscript conclusions are somehow premature.

Response: As suggested by the Editor, we provide more details about the origin and housing of mice in Materials and Methods. “In this study, young male C57BL/6J mice (wild-type mice – WT, n=17, 10-12 weeks old, Velaz, Prague, Czech Republic) and Padi4−/− mice (B6. Cg-Padi4tm1.1Kmow/J, n=15, 10-12 weeks old, Jackson Laboratory, Bar Harbor, Maine, United States, JAX stock #030315) with no prior interventions were used. The Padi4−/− males were generated by mating pairs of homozygous Padi4−/− males and females in the institutional animal facility. Due to small litter sizes, animals used in this experiment originated from several litters and were age-matched. All animals were group-housed (4-5 per cage) in individually ventilated cages (Tecniplast, 1285L, 36.5 x 20.7 x 13 cm), in a room with a controlled light/dark cycle (12/12 hours), constant temperature (22 ± 2°C), and humidity (45-65%). All animals had ad libitum access to standard rodent diet (standard diet for mice, Eypy-KMK20, Bruzovice, Czech Republic) and water.“

Comment: The demonstration of netosis in the tissue is mandatory.

Response: We thank the reviewer for this important comment. To address this point, we evaluated NETosis in kidney tissue by assessing citrullinated histone H3 (citH3) using immunohistochemistry. Citrullination of histone H3 is widely used as a marker associated with NETs formation in tissue sections. In our study, the presence of citH3-positive staining in the renal tissue of adenine-treated animals, regardless of the genotype, indicates activation of the NETosis pathway and supports the occurrence of NETs formation in the kidney during adenine-induced chronic kidney disease. To improve clarity, we have revised the Supplementary methods to explicitly describe the detection of citH3 in renal tissue and its interpretation as evidence of NETs formation. We have also added a sentence in the Discussion highlighting that citH3 is an established marker of NETs formation in experimental models of inflammation and kidney injury.

Comment: Compensatory mechanisms should also be further investigated, knowing the discrepancy between the present study and a previously published one using Pad inhibitor. Generally speaking, the manuscript conclusions are somehow premature.

Response:

We thank the reviewer for this comment. We agree that there is a discrepancy between our findings and a previously published study using a PAD inhibitor. It may be affected by compensatory mechanisms that represent an important aspect of NETosis-related pathways.

In the present study, we focused primarily on the assessment of the role of NETs formation on CKD development in the adenine-induced nephropathy using WT and Padi4−/− mice. However, we acknowledge that inhibition or modulation of PAD4-dependent pathways may trigger compensatory inflammatory mechanisms, including PAD4-independent NET formation pathways. A recent study suggests that hyperuricemia induced by adenine administration can activate a non-canonical pathway of NET formation that is dependent on Gasdermin D (GSDMD) and caspase-11 (1). Similar results were obtained in folic acid-induced nephropathy (2). Importantly, this pathway has been reported to occur independently of the classical PAD4-mediated axis (3). Such mechanisms could potentially explain differences between pharmacological PAD inhibition reported in previous studies and the observations obtained in our experimental model.

To address this point, we have expanded the Discussion section (lines 369–391) to highlight the potential role of compensatory pathways and to discuss possible explanations for the discrepancy with previous PAD inhibitor studies. We also revised the conclusions to avoid overinterpretation and to emphasize that our findings suggest, rather than definitively demonstrate, the involvement of NET-related processes in adenine-induced CKD. Further studies specifically designed to investigate compensatory mechanisms and PAD-independent NET formation will be required to clarify these aspects. These changes have been incorporated into the revised manuscript.

Comment: Thank you for stating in your Funding Statement: “This work was supported by grants from the Grant Agency of Ministry of Education, Science, Research and Sport of the Slovak Republic No. VEGA 1/0212/22 and 1/0801/25 (AGK).” Please provide an amended statement that declares *all* the funding or sources of support (whether external or internal to your organization) received during this study, as detailed online in our guide for authors at http://journals.plos.org/plosone/s/submit-now. Please also include the statement “There was no additional external funding received for this study.” in your updated Funding Statement. Please include your amended Funding Statement within your cover letter. We will change the online submission form on your behalf.

Response: We thank the editor for this suggestion and have updated our Funding Statement accordingly. There was no additional external funding received for this study, and the funders had no role in study design, data collection and analysis, decision to publish, or preparation of the manuscript. We have included this amended Funding Statement in our cover letter as requested.

Amended Funding Statement: This work was supported by the Slovak Research and Development Agency (grants APVV 23 0399 to DVY and APVV-22-0554 to PC) and by the Grant Agency of the Ministry of Education, Research, Development and Youth of the Slovak Republic (grants VEGA 1/0801/25 to AGK, VEGA 1/0297/26 to PV, and VEGA 1/0513/24 to DVY). No additional internal or external funding was received for this study. The funders had no role in study design, data collection and analysis, decision to publish, or preparation of the manuscript.

Comment: Thank you for stating the following financial disclosure: “This work was supported by grants from the Grant Agency of Ministry of Education, Science, Research and Sport of the Slovak Republic No. VEGA 1/0212/22 and 1/0801/25 (AGK).” Please state what role the funders took in the study. If the funders had no role, please state: "The funders had no role in study design, data collection and analysis, decision to publish, or preparation of the manuscript." If this statement is not correct you must amend it as needed. Please include this amended Role of Funder statement in your cover letter; we will change the online submission form on your behalf.

Response: As recommended by the editor, we have corrected the Funding Statement as follows: “This work was supported by the Slovak Research and Development Agency (grants APVV 23 0399 to DVY and APVV-22-0554 to PC) and by the Grant Agency of the Ministry of Education, Research, Development and Youth of the Slovak Republic (grants VEGA 1/0801/25 to AGK, VEGA 1/0297/26 to PV, and VEGA 1/0513/24 to DVY). No additional internal or external funding was received for this study. The funders had no role in study design, data collection and analysis, decision to publish, or preparation of the manuscript.” We included this amended version in the cover letter.

Comment: We notice that your supplementary figures/tables are included in the manuscript file. Please remove them and upload them with the file type 'Supporting Information'. Please ensure that each Supporting Information file has a legend listed in the manuscript after the references list.

Response: We thank the editor for this suggestion. We removed all supplementary figures and tables from the main manuscript file and uploaded them separately as “Supporting Information files”, as requested. Legends for each Supporting Information file have been included in the manuscript following the References section.

Comment: If the reviewer comments include a recommendation to cite specific previously published works, please review and evaluate these publications to determine whether they are relevant and should be cited. There is no requirement to cite these works unless the editor has indicated otherwise.

Response: We did not receive any specific recommendation to cite additional references. However, in the revised manuscript, we have included new references to describe potential compensatory mechanisms that may contribute to the discrepancy between our findings and those of previously published studies using PAD inhibitors (52, 53, 54). In the revised manuscript with track changes, these references are highlighted for ease of review.

Additional Editor Comment: In the study reported in this manuscript, Peter Celec and his team investigated whether neutrophil extracellular traps (NETs) are involved in mouse adenine-induced nephropathy, as a model of chronic kidney diseases. To test this hypothesis, they compared wildtype and Padi4-/- adult mice. They analyzed the renal function and NET-related biomarkers in the plasma and urine of animals during the disease induction. No significant differences were observed between the Padi4-/- and wild-type mice, except a slightly earlier induction of the disease in genetically deficient animals (higher plasma creatinine level and related-body weight in the first days following adenine injections). The authors concluded that NETs are not directly involved in the pathogenesis.

Although this study is of interest, the conclusion seems to be premature.

The main point concerns the presence of NETs in the kidney tissue that was not directly assessed. The authors made the hypothesis that the amounts of NET markers (MPO, NE, extracellular DNA) in the urine/plasma is representative of NETs in the tissue. They must visualized and quantified netosis in the kidney, using citrullinated-histone antibodies. The infiltration of immune cells, including neutrophils, as well.

Response: We thank the editor for this comment. In the present study, we assessed several NET-associated biomarkers in plasma and urine, including MPO, NE, ecDNA, and nucleosomes, to evaluate NET-related activity during adenine-induced nephropathy. To address the editor’s concern regarding immune cell infiltration, neutrophils in kidney tissue were identified by immunostaining using an anti-Ly6G antibody. Following this suggestion, we have now performed quantitative analysis of Ly6G-positive cells, and the corresponding data have been included in the Supporting Information. To provide direct evidence of NETosis in kidney tissue, we additionally assessed citrullinated histone H3 (citH3) by immunohistochemistry. CitH3 is a widely used marker associated with NET formation, reflecting chromatin decondensation during NETosis. Quantification of citH3-positive cells has been performed and the results have been incorporated into the Supporting Information.

Comment: The authors suggested that other Pads may compensate the absence of Pad4, this has to be tested. In particular, Pad inhibitors has to be used in wildtype animals.

In a previously published study, netosis was reported to be associated with adenine-induced nephropathy in mice [47]. This study was consistent with the many different studies showing a role of netosis in human chronic kidney diseases and in other mouse models of the disease; as mentioned by the authors themselves. How do the authors explain this discrepancy? They have to test their hypotheses (see the previous point).

Response: As suggested also in the previous point, we have expanded the Discussion section to better explain the discrepancy between our findings and previously published studies reporting NETosis involvement in adenine-induced CKD. In particular, we discuss the potential role of compensatory mechanisms in Padi4−/− mice, including the activity of other PAD enzymes or activation of PAD-independent NETosis pathways. We highlighted recent evidence suggesting that hyperuricemia induced by adenine administration may trigger a non-canonical NETosis pathway dependent on Gasdermin D and caspase-11, which occurs independently of the classical PAD4-mediated axis. These mechanisms could potentially explain the differences between studies using pharmacological approaches and our genetic PAD4-deficient model.

While we agree that the use of PAD inhibitors in wild-type animals could provide valuable mechanistic insight, conducting such experiments would require additional resources and funding, which was beyond the scope of the present study. We have therefore noted this as a potential direction for future research in the Discussion.

Minor comments: What is the genetic background of Padi4-/- mice?

Response: As recommended by the editor, we included the background of Padi4−/− mice in Materials and Methods. “In this study, young male C57BL/6J mice (wild-type mice – WT, n=17, 10-12 weeks old, Velaz, Prague, Czech Republic) and Padi4−/− mice (B6. Cg-Padi4tm1.1Kmow/J, n=15, 10-12 weeks old, Jackson Laboratory, Bar Harbor, Maine, United States, JAX stock #030315) with no prior interventions were used.”

What is the response of female mice?

Response: In the present study, we included only male mice to reduce variability related to sex-specific differences in hormone levels and disease susceptibility. The impact of Padi4 deficiency on adenine-induced CKD in female mice was not investigated and remains to be elucidated. We have added this limitation to the Discussion and noted that future studies should include female animals to determine whether sex differences influence NETs formation and kidney injury in this model.

Please give scale bars on (immuno)histological pictures, rather than magnifications.

Response: We thank the reviewer for this suggestion. All (immuno)histological images have been updated to include scale bars instead of magnification information.

Please give the origin of antibodies.

Response: We thank the reviewer for this comment. The origin (manufacturer, catalog number) of all antibodies used in the study has now been added to the Supporting Information file.

The rise in plasma ecDNA was higher in Padi4-/- as compared to wild-type mice, but the difference was not statistically significant. Did the authors consider to increase the number of analyzed mice? For other non-statistically significant different values as well.

Response: We thank the reviewer for this important comment. We agree that increasing the number of animals could potentially improve the statistical power and help clarify subtle differences between the groups. However, the number of animals used in the present study was determined in accordance with ethical considerations for animal experimentation and is comparable to that used in similar experimental studies investigating adenine-induced CKD. Increasing the sample size would require additional animal experiments that were beyond the scope of the present study.

Importantly, the lack of statistically significant differences between wild-type and Padi4−/− mice was consistent across multiple parameters assessed in our study, suggesting that PAD4 deficiency does not have a major impact on disease progression in this model.

Reviewer 1: Kovalčíková et al. present a manuscript where they use PAD4-deficient animals to investigate how NETs can potentially contribute to chronic kidney disease in a model of adenine-induced nephropathy. They find that WT and PAD4-/- mice have comparable responses, except an earlier exacerbated response in PAD4-/- animals.

Comment

---

## [Decision Letter · Decision Letter 1]

7 May 2026

Neutrophil extracellular traps in the animal model of adenine-induced chronic kidney disease

PONE-D-25-62916R1

Dear Dr. Celec,

We’re pleased to inform you that your manuscript has been judged scientifically suitable for publication and will be formally accepted for publication once it meets all outstanding technical requirements.

Kind regards,

Michel Simon, Ph. D.

Academic Editor

PLOS One

Additional Editor Comments (optional):

All concerns have been addressed. Thank you.

Reviewers' comments:

Reviewer's Responses to Questions

**Comments to the Author**

1. If the authors have adequately addressed your comments raised in a previous round of review and you feel that this manuscript is now acceptable for publication, you may indicate that here to bypass the “Comments to the Author” section, enter your conflict of interest statement in the “Confidential to Editor” section, and submit your "Accept" recommendation.

Reviewer #1: All comments have been addressed

2. Is the manuscript technically sound, and do the data support the conclusions?

Reviewer #1: Yes

3. Has the statistical analysis been performed appropriately and rigorously? 

Reviewer #1: Yes

4. Have the authors made all data underlying the findings in their manuscript fully available?

Reviewer #1: Yes

5. Is the manuscript presented in an intelligible fashion and written in standard English?

Reviewer #1: Yes

6. Review Comments to the Author

Reviewer #1: All of my comments have been addressed in the revised manuscript.

7. PLOS authors have the option to publish the peer review history of their article (what does this mean?). If published, this will include your full peer review and any attached files.

Reviewer #1: **Yes:** Kimberly Martinod

---

## [Editor Report · Acceptance letter]

PONE-D-25-62916R1

PLOS One

Dear Dr. Celec,

I'm pleased to inform you that your manuscript has been deemed suitable for publication in PLOS One. Congratulations! Your manuscript is now being handed over to our production team.

Kind regards,

on behalf of

Dr. Michel Simon

Academic Editor

PLOS One